# Synthesis of Natural Nano-Hydroxyapatite from Snail Shells and Its Biological Activity: Antimicrobial, Antibiofilm, and Biocompatibility

**DOI:** 10.3390/membranes12040408

**Published:** 2022-04-08

**Authors:** Hanaa Y. Ahmed, Nesreen Safwat, Reda Shehata, Eman Hillal Althubaiti, Sayed Kareem, Ahmed Atef, Sameer H. Qari, Amani H. Aljahani, Areej Suliman Al-Meshal, Mahmoud Youssef, Rokayya Sami

**Affiliations:** 1The Regional Center for Mycology and Biotechnology, Al-Azhar University, Cairo 11787, Egypt; hanaa_hyk@yahoo.com (H.Y.A.); nesrrcmb@hotmail.com (N.S.); dredash81@gmail.com (R.S.); dr.sayedkareim@gmail.com (S.K.); ahmed4ui2@yahoo.com (A.A.); 2Department of Biotechnology, Faculty of Science, Taif University, P.O. Box 11099, Taif 21944, Saudi Arabia; i.althubaiti@tu.edu.sa; 3Department of Biology, Al-Jumum University College, Umm Al-Qura University, Makkah 21955, Saudi Arabia; shqari@uqu.edu.sa; 4Department of Physical Sport Science, College of Education, Princess Nourah bint Abdulrahman University, P.O. Box 84428, Riyadh 11671, Saudi Arabia; ahaljahani@pnu.edu.sa; 5Department of Biology, College of Science and Humanities in Al-Kharj, Prince Sattam bin Abdulaziz University, Al-Kharj 11942, Saudi Arabia; a.almashal@psau.edu.sa; 6Food Science and Technology Department, Faculty of Agriculture, Al-Azhar University, Cairo 11787, Egypt; 7Department of Food Science and Nutrition, College of Sciences, Taif University, P.O. Box 11099, Taif 21944, Saudi Arabia

**Keywords:** snail shell, hydroxyapatite, nanoparticles, antimicrobial, antibiofilm, biocompatibility

## Abstract

Hydroxyapatite nanoparticles (HAn) have been produced as biomaterial from biowaste, especially snail shells (*Atactodea glabrata)*. It is critical to recycle the waste product in a biomedical application to overcome antibiotic resistance as well as biocompatibility with normal tissues. Moreover, EDX, TEM, and FT-IR analyses have been used to characterize snail shells and HAn. The particle size of HAn is about 15.22 nm. Furthermore, higher inhibitory activity was observed from HAn than the reference compounds against all tested organisms. The synthesized HAn has shown the lowest MIC values of about 7.8, 0.97, 3.9, 0.97, and 25 µg/mL for *S. aureus*, *B. subtilis*, *K. pneumonia*, *C. albicans*, and *E. coli*, respectively. In addition, the HAn displayed potent antibiofilm against *S. aureus* and *B. subtilis*. According to the MTT, snail shell and HAn had a minor influence on the viability of HFS-4 cells. Consequently, it could be concluded that some components of waste, such as snail shells, have economic value and can be recycled as a source of CaO to produce HAn, which is a promising candidate material for biomedical applications.

## 1. Introduction

Since humans used antibiotics as treatment in their early history, antibiotic resistance has arisen as problematic. It currently represents a huge danger to human health [1]. Pathogenic bacteria exist most regularly in the biofilm form responsible for this problem, making extra bacterial resistance to antimicrobial medicines. Biofilm is thought to be an essential driver of antibiotic-resistant bacteria. The high resistance of biofilms to current antimicrobial treatments a challenge. Likewise, biofilm eradication is challenging, regardless of whether in medication or industry. Antibiotic treatment alone regularly neglects to destroy microbial biofilms [2,3,4]. Microorganisms have developed a variety of defense mechanisms to maintain activity in the face of human immune responses and antibiotic therapy. Biofilms are microbes that adhere to biotic or abiotic surfaces and are surrounded by an extracellular polymeric substance (EPS) [4,5]. So, developing new antimicrobial therapies that can overcome this resistance is urgent. Natural sources are considered an alternative to antibiotics and have fewer side effects than allopathic cures [6,7,8,9].

Molluscs are considered a natural source for pharmacological applications. Furthermore, mollusc aquaculture is a highly cost-effective food source that has the potential to play a significant role in global food security in the future [10,11]. With creation expanding worldwide, the time has come to assess all aspects of aquaculture while considering its expanding role as a food source. Snails belong to the phylum molluscs and gastropods. This class incorporates gastropods, slugs, and snails [12].

One element of mollusc aquaculture waste creation that is sometimes ignored is the formation of calcareous shells. Shell trash from the aquaculture industry is commonly recognized as a nuisance waste product; shell debris may pose a substantial logistical and financial challenge for shellfish growers, sellers, and consumers. Species subordinate shells can account for up to 75% of an organism’s total weight [13]. Shells are regarded as a useful biomaterial by the aquaculture and fishing industries. They may be reused for environmental and economic reasons.

Hydroxyapatite is produced from a variety of sources, including eggshells [14,15] animal bones [16], seashells [17], and plants [18]. These resources are a good source of hydroxyapatite or a viable option for Ca and P precursors in producing pure and thermally stable hydroxyapatite. HA made from natural sources or waste might be more helpful since it frequently includes essential ions present in biological HA [19].

When crustaceans are eaten, much calcium- and HA-rich waste is produced. So, different bioactive compounds can be prepared from marine waste [19]. With the increasing significance of HA in various clinical fields, many overview studies on the synthesis of HA nanoparticles and their usage and production methods have been authored. Nonetheless, no critical studies on the composition of hydroxyapatite nanoparticles from biowaste products or other sources have been conducted so far [19].

Emerging bioceramic materials, which are widely employed in numerous biomedical applications, including dentistry, are biologically relevant forms of CaPs. They offer excellent biomedical and biological characteristics.

CaPs have unique properties because their composition and structure are identical to human teeth and bones. As a result, CaPs possess excellent possessions. Biocompatibility and one-of-a-kind bioactivity are two terms that are frequently used [20].

The most common type of CaPs used in dental applications appears to be apatites (HA, CDHA, and FA). Because nano-dimensional and nano-crystalline apatites are often regarded as model compounds for dental enamel due to chemical and phase similarities [21], their use in restorative dentistry has a number of promising benefits, including intrinsic radio-opaque response, enhanced polishability, and improved wear performance. They also have a hardness that is comparable to that of actual teeth [22].

Nano-dimensional HA particles, for example, were discovered to have the capacity to permeate a dentin collagen matrix that had been demineralized, which provides a suitable scaffold.

Teeth caries are produced by bacteria -creating acid in biofilms on dental surfaces; as such, avoiding it requires control of the microorganisms that produce the acids. Mouth rinses containing HA were reported to reduce early bacterial colonization [23]. In a separate study, CDHA-osteopontin biocomposite particles were designed to bind to bacteria in biofilms, prevent biofilm formation without killing the microflora, and release orthophosphate ions to buffer bacterial acid production when the pH fell below six. According to the findings, treatment with either CDHA-osteopontin or pure osteopontin resulted in reduced biofilm development than untreated controls. Thus, osteopontin was responsible for the anti-biofilm effect of the CDHA-osteopontin particles, while CDHA was responsible for the buffering effect, which kept pH constant [24].

Coatings of CaPs (both HA and -TCP) have been effectively applied to titanium implants, and the coated implants that performed well were discovered to be suitable for use as anchoring in short-term orthodontics. Both types of coatings appeared to be effective stimulators of new bone formation [25].In addition, researchers added nano-dimensional composite particles of HA with silver to commercially made light cure adhesive Transbond XT (3M Oral Care, St. Paul, MN, USA) to boost antibacterial capabilities. Bacterial growth inhibition zones were observed on Transbond XT composite discs containing 5% and 10% Ag/HA, and antibacterial capabilities against biofilms were discovered [26].

The present study aims to recycle the waste product of snail shells in a medical application. So, we use green, economical, and waste-derived HAn prepared from snail shells (*Atactodea glabrata*) as a cheap, readily available, non-toxic means of producing antimicrobial and anti-biofilm agents to overcome antibiotic resistance as well as biocompatibility with normal tissues.

## 2. Materials and Methods

### 2.1. Snail Collection

Snails were collected from Ras Sidr, South Sinai Governorate, Egypt. The samples were washed and cleaned with water to remove sand and other dust particles [27].

### 2.2. Identification of Snails

Snail samples were verified according to the whole number and the opening position [28].

### 2.3. Preparation of Snail Powder

The clean snails were dried in the oven at 100 °C for 24 h before being crushed using milling balls. The powder was placed in an electrical furnace and heated to 900 °C for three hours to convert the CaCO_3_ in the seashell to CaO. XRD was used to establish the presence of CaO in the resultant powder [29]. By adding the predetermined amount of distilled water to the resulting powder, 1 M of calcium hydroxide solution was created. The solution was agitated for about two hours to achieve homogeneous Ca(OH)_2_ mixing.

### 2.4. Preparation of Hydroxyapatite from Snail

Synthesis of hydroxyapatite was conducted by using calcination and wet chemical precipitation technique. Briefly, phosphoric acid (0.6 M, H_3_PO_4_) was dropped to Ca(OH)_2_ at a 15–20 drop/min rate with continuous stirring at room temperature to produce hydroxyapatite. Then adding ammonium hydroxide solution (NH_4_OH) until the reaction was completed, the pH was kept around pH 8 for around 2 h, and the solution was constantly stirred and aged. The gelatinous white precipitate emerged after the solution stopped stirring and was left to precipitate overnight. The solution was then filtered and rinsed many times with distilled water before being dried at 200 °C for 24 h to eliminate the water. Finally, the hydroxyapatite obtained was sintered in the furnace at a temperature of 1000 °C for 4 h. The resulting hydroxyapatite was characterized using XRD, EDX, and FT-IR [30,31].

### 2.5. Physical Characterization

#### 2.5.1. FTIR Spectrum

FTIR spectrum was performed using an FTIR spectrophotometer (Model-4100, Jasco, Easton, MD, USA) to characterize the snail shell powder and hydroxyapatite. FTIR spectra were recorded in the range of 400–4000 cm^−1^ [32].

#### 2.5.2. X-ray Diffraction

The hydroxyapatite samples’ crystalline phases were identified using the X-ray powder diffractometer (XRD, PW3040/60, PANalytical, Almelo, The Netherlands). A Cu-Kα radiation light (λ = 1.5401 Å), with power generated at 30 mA and 45 kV with a scan speed of 2°/min over a 2θ-range of 5–75° was used. The diffraction peaks of the examined crystalline phase were studied with (JCPDS, 896438, Karlsruhe, Germany) files and were marked with different symbols.

#### 2.5.3. Energy Dispersive Analysis of X-ray (EDX)

The presence of Calcium and phosphate elements was confirmed through EDX. The X-ray micro-analyzer carried out the EDX microanalysis (Oxford 6587 INCA, Oxford Instruments, Abingdon, UK) connected with the JEOL JSM-5500 LV-SEM (Jeol, Tokyo, Japan) at 20 kV [33].

#### 2.5.4. TEM

For TEM analysis, a drop of the solution was placed on the carbon-coated copper grids and dried by allowing water to evaporate at ambient temperature. Electron micrographs were taken using a JEOL JEM-1010-TEM (Jeol, Tokyo, Japan) at 70 kV [34].

### 2.6. Antimicrobial Activity

Microorganism cultures are used in this study, which includes gram-positive bacteria, *Staphylococcus aureus* (ATCC 6538), and *Bacillus sutilis* (ATCC 6633) as well as gram-negative bacteria, *Klebsiella pneumonia* (ATCC 13883), and *Escherichia coli* (ATCC 8739), unicellular fungi, *Candida albicans* (ATCC 10221), and filamentous fungi, *Aspergillus Niger*. Nutrient agar plates for bacteria and malt agar plates for fungi were prepared. Then, 0.1 mL of the inoculum from the standardized culture of the test organism spread uniformly. Wells were made using a sterile tool of diameter 10 mm and 0.1 mL of each sample. A standard antibiotic was added to each well separately. Gentamicin, a common antibiotic, was tested against bacteria, whereas amphotericin B was tested against fungus. The plates were incubated for 24 h at 37 °C. Antimicrobial activity was expressed as the diameter of the inhibition zone [35].

### 2.7. Minimal Inhibitory Concentration (MIC)

The MIC of the samples was determined using sterile 96-well plates. In brief, 0.1 mL of standardized inoculums of 1 to 2 × 10^5^ CFU/mL for *E. coli* (ATCC 8739) and *S. aureus* (ATCC 6538) was added to each well. For the stock solution preparation, 10 mg of each sample was dissolved in 10 mL of sterile distilled water (1 mg/mL). Serial two-fold dilutions of the sample were added into a 96-well tissue culture plate. The plate was incubated at 37 °C for 24 h. The MIC was detected by measuring optical density at 600 nm using a microplate reader (Sunrise, TECAN Inc., San Jose, CA, USA) [36].

### 2.8. Antibiofilm Activity

Single-species biofilms of *E. coli* (ATCC 8739) were formed by adding 10 μL of the bacterial cell suspension to 0.19 mL TSB media in each well, and 0.2 mL of sterile distilled water was added in wells to reduce the water loss. Then the microtiter plate was incubated overnight at 37 °C. The following day, the wells were rinsed twice with 0.9% NaCl; after that, the fresh medium containing the samples at their MIC concentration was added; a 0.9% NaCl solution was used as a control. After incubation for another 24 h, the medium was removed. Each well was gently washed three times with phosphate buffer saline or sterile saline water and left to dry at room temperature. Then, 200 μL of crystal violet solution (0.2%) was added to all wells. After 15 min, the excess stain was dragged, and plates were washed twice and air-dried. Finally, the cell-bound crystal violet was resuspended in 33% acetic acid. The biofilm growth was monitored at 600 nm using a microplate reader (Sunrise, TECAN Inc., San Jose, CA, USA) [37].
(1)$$\mathrm{Biofilm}\;\mathrm{inhibition}\;\mathrm{ability}\;\mathrm{of}\;\mathrm{sample}=\frac{\left(\mathrm{Absorb}.\mathrm{control}-\mathrm{Absorb}.\mathrm{blank}\right)-\left(\mathrm{Absorb}.\mathrm{sample}-\mathrm{Absorb}.\mathrm{blank}\right)}{\left(\mathrm{Absorb}.\mathrm{control}-\mathrm{Absorb}.\mathrm{blank}\right)}\times 100$$

### 2.9. MTT Assay

The MTT test was used in triplicate to evaluate the viability of control and treatment cells. The MTT assay is a laboratory test and a widely used colourimetric assay (one that detects colour changes) for determining cellular proliferation. The cell lines used in this work, i.e., HSF-4 Skin (normal fibroblast cell line), were obtained from the American Type Culture Collection (Manassas, VA, USA). HSF-4 cells were seeded at a density of 1 × 10^4^ cells/well in 96-well plates with 100 µL of growth media. Cells were allowed to attach for 24 h until conjunction, and then washed with a phosphate buffer solution before being treated with various concentrations of samples in a new maintenance medium ranging from 500 to 15.63 µg. They were then incubated for 24 h at 37 °C. A 96-well plate was loaded with serial two-fold dilutions of the material using a multichannel pipette (Eppendorf, Germany). After treatment, the culture supernatant was replaced with a new medium (24 h). The cells in each well were then treated for 4 h at 37 °C with 0.1 mL of MTT solution (5 mg/mL). The MTT solution was withdrawn once the incubation period was completed, and 0.1 mL of DMSO was added to each well. A microplate reader (Sunrise, TECAN Inc. USA) was used to measure the absorbance at 570 nm [38].

### 2.10. Microscopic Studies

An inverted microscope (CKX41; Olympus, Tokyo, Japan) coupled with a digital camera was used to acquire pictures showing the morphological alterations of the treated cells stained with 0.25% crystal violet compared to control cells.

### 2.11. Statistical Analysis

Experiments were carried out three times in total. The mean ± standard deviation (SD) is used to represent all data.

## 3. Results

### 3.1. Identification of the Snail

The snails have been described as *Atactodea glabrata* Family: Mesodesmatidae based on their characteristics of morphology and physiology, Figure 1.

### 3.2. Characterization of Snail Shell Powder and HAn

#### 3.2.1. FT-IR Analysis

The FTIR spectra of raw snail shells powder and HAn formed via a chemical precipitation method using thermal decomposition (calcite) to the snail shells are shown in Figure 2a. The *Atactodea glabrata* snail shell powder spectrum as a calcium source has been confirmed by identifying the main functional groups of the snail in the region between 400 cm^−1^ and about 1500 cm^−1^. The spectrum values in Figure 2a were compared to the previous studies [39], and the data support the presence of CaCO_3_. The snail shell powder shows (Figure 2a) characteristic absorption bands for carbonate ions (CO_3_) at a spectrum of 1448.15 cm^−1^, 860.60 cm^−1^, and 712.66 cm^−1^, respectively. In addition, the major vibration CO_3_ bonds in aragonite occur at a wavenumber of 1448.15 cm^−1^. Aragonite is naturally associated with ores and the 1082.85 wavenumber corresponds to sulfate [26].

Figure 2b describes the FT-IR spectra of HAn designed by a chemical precipitation procedure using calcite from the snail shells. New regions have appeared in the FT-IR spectra of HAn, which occurred due to replacing carbonate (CO_3_) with phosphate (PO_4_) in calcite. The specific bands for PO_3_ were recorded from 1100 to 1027 cm^−1^, from 970 to 934 cm^−1^, from 575 to 526 cm^−1^, and from 493 to 430 cm^−1^, inferring the composition of HAn, and in agreement with published data on HA [40]. Furthermore, the original band for functional groups CO_3_ and OH was indicated at 724 cm^−1^ and 634 cm^−1^ for HA. In addition, the presence of PO_4_ groups, characteristic of β-TCP, is shown by absorption bands from 1100 to 1155 cm^−1^, which is characteristic of [PO_4_] _v2_ group (_v2_ O-P-O) bending variations [41,42,43].These results have confirmed the composition of HAn and have also been supported by XRD data. Thus, our outcomes agree with previously published data on HA [39].

#### 3.2.2. XRD Analysis

The hydroxyapatite samples’ crystalline phases were identified using the powder X-ray diffraction. The results of XRD patterns for the phases of calcium phosphate present in the samples prepared are presented in Figure 3. The XRD pattern showed that the composite was mainly constituted of Hydroxyapatite mixed with a phase of tricalcium phosphate. The XRD presented in the figures was compared with JCPDS (896438) [33], showing that it was formed of phase clear crystalline HA with a hexagonal crystal structure [33,34].

#### 3.2.3. Energy Dispersive X-ray (EDX) Analysis

The EDX profile of snail shell and HAn is shown in Table 1. The quantitative and qualitative status of components implicated in the snail shell and HAn is determined using EDX. EDX micro-analysis measured the energy and intensity distribution of X-ray signals generated by a focused electron beam on an object. It is clear that the percentages of Ca, O, and P were 35.54%, 61.24%, and 0.01%, respectively, for the snail shell of *Atactodea glabrata*. While the percentages of Ca, O, and P for HAn are about 24.53%, 48.48%, and 20.67%, respectively (Table 1). According to the findings, the average proportion of calcium in snail shell powders for *Atactodea glabrata* was 35.54%.

The findings also showed that the investigated snail shell, which contains the bulk of calcium carbonate, may be utilized as a calcium precursor in the production of calcium phosphate (Ca_3_(PO_4_)_2_) bioceramics. Calcium phosphate-based biomaterials are a class of substances with a Ca/P molar ratio of 0.5–2 [44,45,46,47]. They are commonly used as biomaterials for the reconstruction of various bone defects, notably in dentistry and orthopedics.

The prior finding that the prepared HA had a Ca/P molar ratio in the range of 0.5–2 is consistent with our results in which a Ca/P molar ratio of about 1.19. Our findings support those of [48,49,50], who determined that the snail shell contains enough calcium carbonate to be used as a calcium source (Ca). In dry matter, the calcium concentration of snail shells is usually around 34–35%. For *Achatina achatina* and *Lanistevaricus* snails, the average proportion of CaCO_3_ in shell powders was 98.5% and 98.75%, respectively, according to Osseni et al. [51]. The greatest Ca^2+^ and Fe^2+^ concentrations were found in *M. lussoria* and *M. mercenaria* (clam species), while the lowest amounts were found in *A. achatina* [52].

#### 3.2.4. TEM

The data obtained from the transmission electron-micrograph showed the distinct shape of snail shell powder of *Atactodea glabrata* and HAn (Figure 4). The particles with 13.302 and 15.22 nm diameters for *Atactodea glabrata* snail shell and HAn powders, respectively. According to the previous studies, nano-sized (20 nm) HA, which were counterparts to the fundamental building blocks of the enamel rods, could perform a localised biomimetic repair of the enamel surface. Because of these similarities, artificial biomaterials could adhere well to natural tissues. Furthermore, because secondary caries were controlled and hardness was maintained, the enamel structure was strengthened by nano-sized HA [25]. Dentin remineralization may occur in a remineralizing environment [53]. Furthermore, it was shown that nano-sized HA particles may self-assemble into enamel-like patterns [54].

### 3.3. Antimicrobial Activity of Atactodea glabrata Snail Shell

Most of the literature concentrates on the flesh and haemolymph of molluscs, whereas the shell, particularly the traditional usage of mollusc shells, receives very little attention. Snail shell powders were tested for antibacterial activity in this study.

Snail shell powder of *Atactodea glabrata* was tested for antimicrobial using the agar well diffusion technique. Table 2 displays the sample’s inhibition zone diameter (mm) against various pathogenic organisms. The outcomes revealed that the snail shell powder of *Atactodea glabrata* showed a higher inhibitory effect than the reference compound gentamicin against *B. subtilis* (32 mm) Figure 5. In addition, high inhibitory activity was detected against *S. aureus* (20 mm) and *E. coli* (26 mm), which wasstill fewer than the reference compound. Similarly, moderate inhibitory activity was observed against *C. albicans* (17 mm). In contrast, no inhibitory effect against *A. niger* and *K. pneumonia* occurred. Our findings contradict earlier findings, indicating that mollusc shells do not have antibacterial properties since they did not prevent pathogenic isolates from growing. According to Kehinde et al. [52], snail haemolymph did not prevent the development of fungal and bacterial isolates. The epiphragm of albino and normal skinned *A. marginata*, conversely, inhibited four bacterial isolates better than conventional antibiotics (streptomycin) [55].

### 3.4. Antimicrobial of HAn

Previous results found that the snail shell powder of *Atactodea glabrata* had high inhibitory activity. The hydroxyapatite was prepared from the *Atactodea glabrata* shell according to these results. Elevated inhibitory activity occurred from HAn against all tested organisms. Table 3 showed higher inhibitory effect than the reference compound gentamicin against *B. subtilis* (43 mm), *S. aureus* (30 mm), and *K. pneumonia* (42 mm). In contrast, a slight inhibitory effect was observed against *E. coli* (26 mm). In addition, a higher inhibitory effect than the reference compound amphotericin B against *C. albicans* (44 mm) and *A. niger* (22 mm) was observed (Figure 6).

According to Osseni et al. [51], only calcium phosphate bioceramics produced from *Lanistes varicus* snail shells [56] may be utilized to prevent the formation of microorganisms that cause tooth cavities. As a bioactive biomaterial in dental applications, its antibacterial properties should be exploited to their full potential [57]. Products made from *Achatina achatina* snail shell powder, conversely, showed no antibacterial action on both types of bacteria.

### 3.5. MIC of HA

The MIC of HAn with strong antibacterial activity was investigated further by a tube dilution technique against *S. aureus, B. subtilis, E. coli, K. pneumonia*, and *C. albicans* (Table 4). The HAn extracted from the snail shell *Atactodea glabrata* has shown MIC values of 7.8, 0.97, 3.9, and 0.97 mg/mL for *S. aureus, B. subtilis, K. pneumonia*, and *C. albicans*, respectively, confirming its antibacterial effectiveness. In contrast, HAn showed a 25 µg/mL MIC for *E. coli*. The highest activity of the extracted HAn can be explained by the presence of slightly basic compounds (HA, TCP) to some extent neutralizes the acid molecules, provides with a weak pH-buffering effect at the polymer surface and, therefore, reduce the bacterial growth in which bacteria creating acid in biofilms on dental surfaces [56,58,59].

### 3.6. Biofilm Formation Effects of HAn

Biofilms are crucial virulence agents for certain pathogen microorganisms, and some biofilm infections appear almost impossible to stop [60]. Pathogens such as *P. aeruginosa* [61], *S. epidermidis* [62], *C. albicans* [63], *S. aureus* [64], and *S. enterica* [65] can form biofilms. These pathogens form biofilms in the same way and share many characteristics [66,67].

The HAn extracted from the snail shell of *Atactodea glabrata* was assessed for its antibiofilm activity against the selected bacteria (*S. aureus*, *B. subtilis*, *E. coli*, and *C. albicans*) using crystal violet stain. Results in Figure 7 showed that the HAn exhibited a strong biofilm inhibition effect opposed to *S. aureus* and *B. subtilis*, with a biofilm inhibitory ratio of 81.26% and 77.25%, respectively. At the same time, it showed moderate biofilm inhibition activity against *E. coli* 58.7% and *C. Albicans* 51.2%. According to Abdelraof et al. [68], the HAn had a significant biofilm inhibition effect against *B. subtilis*, a biofilm inhibitory ratio of 77.25%, and a mild biofilm inhibition effect against *E. coli*, with a biofilm inhibitory ratio of 66.69%.

### 3.7. MTT Assay

The in vitro biocompatibility of materials was examined in this study by using an MTT assay against normal skin cells (HSF-4). The cytotoxic effects of samples after 24 h of treatment on the HFS-4 cell line are shown in Figure 8a. The results suggested that the snail shell powder, *Atactodea glabrata*, slightly affected the cells of HFS-4 even at a high concentration, with a viability percent of about 96.43% at 125 µg/mL. No visible cytotoxic effects were noted against HSF-4 cells, rising dilutions from 125 to 15.63 µg/mL. Moreover, the HAn displayed viability of about 76.83 at 125 µg/mL. No cytotoxic effects were detected against normal cells with rising dilutions from 62.5 to 15.63 µg/mL.

The reverse microscope examination revealed a low cytotoxic effect of all treatments after 24 h in the culture at 125 µg/mL. At confluence, control HFS-4 cells established a monolayer of elongated cell sin Figure 8b. After treatment with 125 µg/mL of both samples, only minor morphological alterations were found. With higher dilutions, the number of floating cells was reduced after 24 h. As an untreated cell, the HFS-4 cells were able to multiply and develop in a monolayer. Previous research supports the findings, which indicate that the primary objective of biomaterials is to promote cell development while also protecting them from harmful consequences [68,69,70].

Compared to commercially available HA, HA isolated from tilapia (*Oreochromis* sp.) significantly improved MG63 type cell viability. The smaller particle size (719.8 nm) is credited with the increased biological activity [19]. At the same time, the HA sample retains good biocompatibility at neutral pH values of the medium, which is the best pH for cell growth [71].

pH variations influenced the physicochemical and biological characteristics of the samples. It has been shown that NBG-CaCl_2_ offers normal cells with somewhat higher viability than NBG; this might be related to Ca^2+^, which plays a function in biological system control [68]. Due to adhesion of molecules on cell surfaces, studies have revealed that appropriate Ca^2+^ and Mg^2+^ ions can enhance cell attachment and proliferation [72,73,74]. According to several research studies, nanosized HA particles improved the mechanical and biological characteristics of the scaffold by increasing protein adsorption and cell adherence to the interior surfaces [75].

## 4. Conclusions

A green, eco-friendly, and biocompatible waste-derived HAn was prepared from snail shells, *Atactodea glabrata*. TEM showed the particle size of prepared snail shell powder of *Atactodea glabrata*, and HAn powders ranging from 13rangingfrom13.3 to15.2 nm in diameter. Snail shell powder has high antimicrobial activity. The HAn prepared from the snail shell *Atactodea glabrata* presented a higher inhibitory effect than the standard compounds against all tested organisms. The lowest MIC values were observed against *E. coli*, *S. aureus*, *B. subtilis*, *K. pneumonia*, and *C. albicans* from HA. In addition, the HAn displayed potent antibiofilm against *S. aureus* and *B. subtilis*. At the same time, it had moderate biofilm inhibition activity against *E. coli* and *C. Albicans*. The findings observed that the snail shell powder, *Atactodea glabrata*, slightly affected the viable cells of HFS-4 even at higher concentrations, followed by HAn. In conclusion, HAn prepared from snail shells is expected to be a cheap, natural, and biocompatible material. The HAn is promising for producing antimicrobial and anti-biofilm agents to overcome multidrug-resistance bacteria.

## Figures and Tables

**Figure 1 membranes-12-00408-f001:**
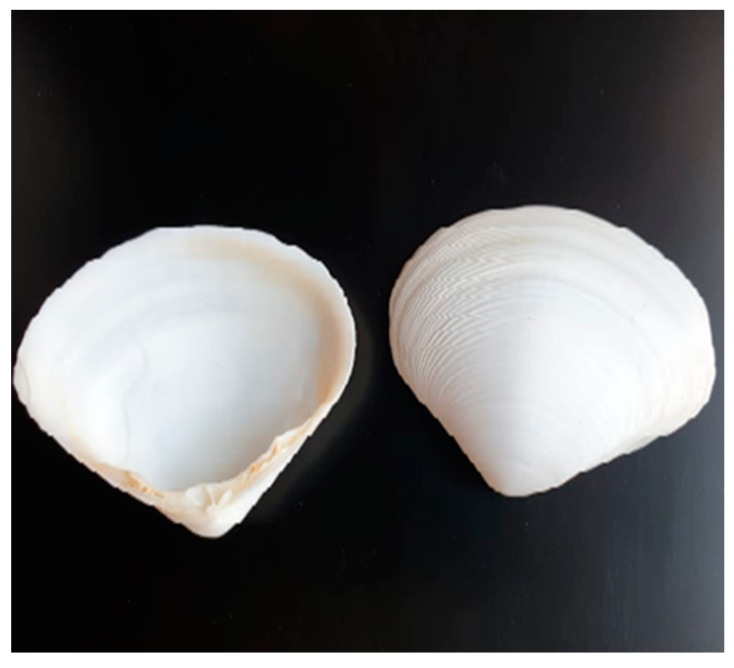
Species of gastropod snails and shells (*Atactodea glabrata)* collected from Ras Sidr, South Sinai Governorate, Egypt: Family Mesodesmatidae.

**Figure 2 membranes-12-00408-f002:**
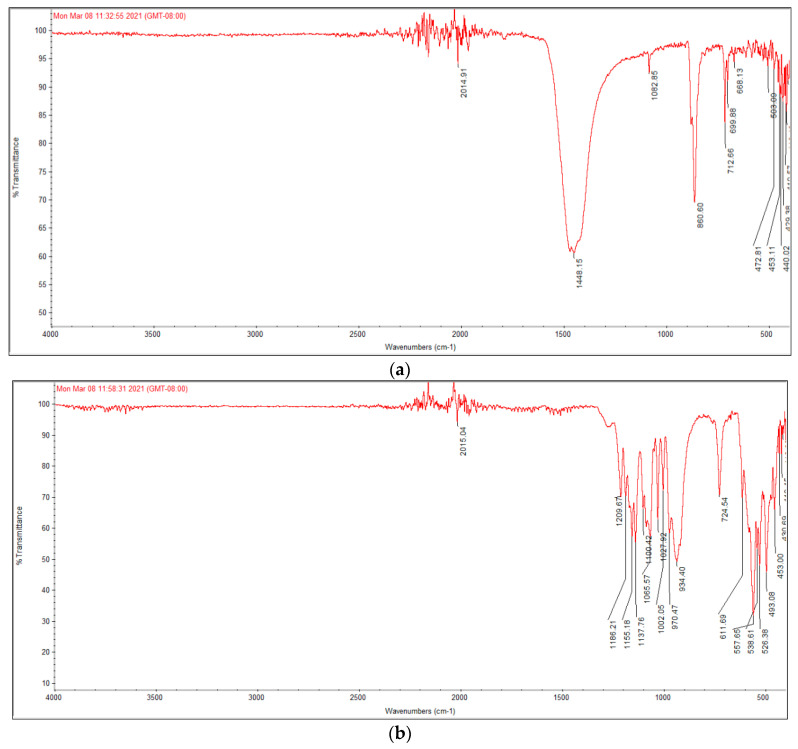
FT-IR of (**a**) *Atactodea glabrata* snail shell, and (**b**) HAn.

**Figure 3 membranes-12-00408-f003:**
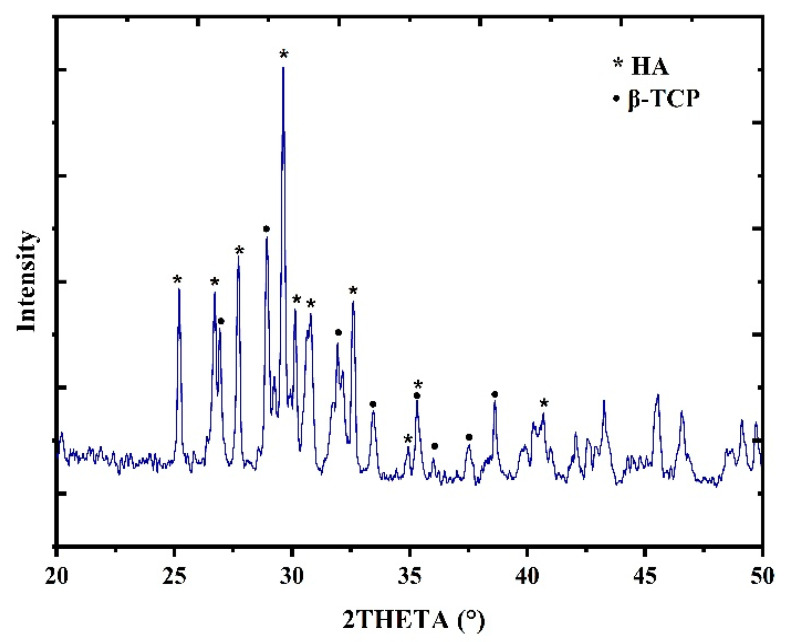
XRD of HAn prepared from snail shell of *Atactodea glabrata*.

**Figure 4 membranes-12-00408-f004:**
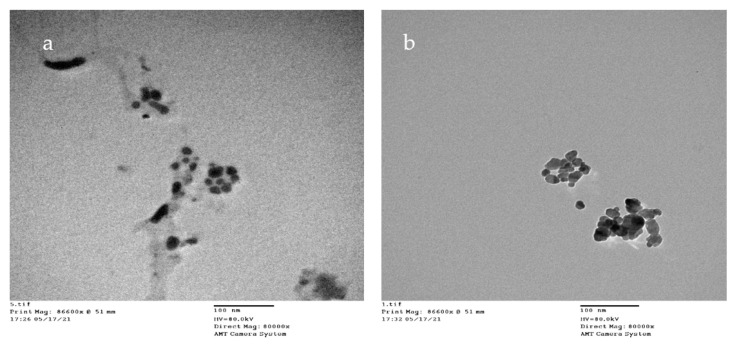
TEM of (**a**) HAn, and (**b**) snail shell of *Atactodea glabrata*.

**Figure 5 membranes-12-00408-f005:**
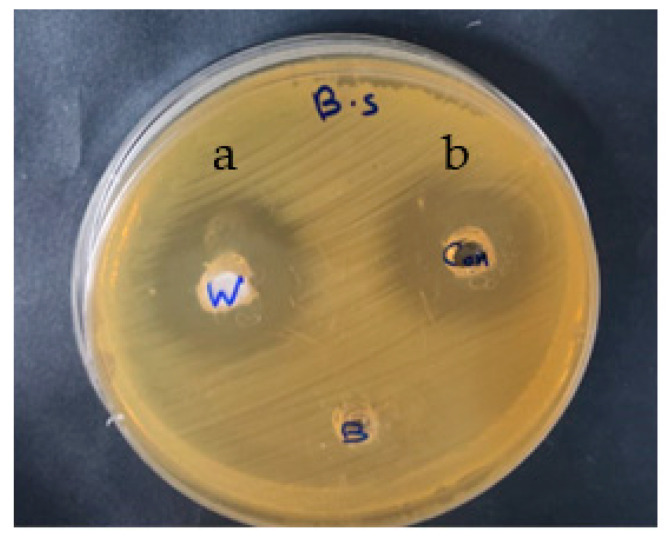
Antimicrobial activity of Snail shell against *B. subtilis* (B.S) (**a**) Snail shell powder (labeled as w) (**b**) Gentamicin (labeled as con).

**Figure 6 membranes-12-00408-f006:**
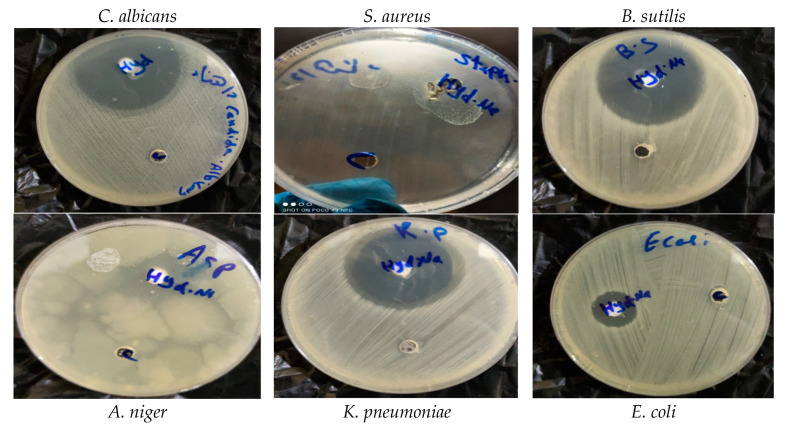
Antimicrobial activities of HA by agar well diffusion method.

**Figure 7 membranes-12-00408-f007:**
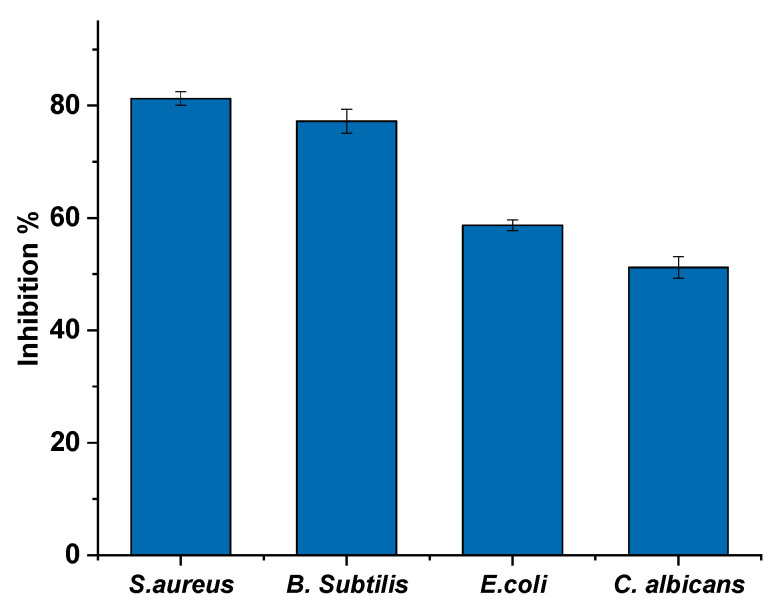
Antibiofilm activity of HAn.

**Figure 8 membranes-12-00408-f008:**
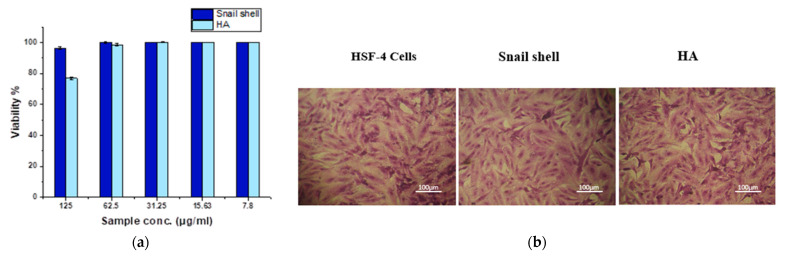
(**a**) The cytotoxic effects of samples with concentrations (15.63, 31.25, 62.5, and 125 µg/mL) after 24 h treatment against HSF-4 cells using MTT assay. (**b**) Morphological features of HSF-4 cells treated with 125 µg/mL of samples showed after 24 h treatment under an inverted microscope compared to control HSF-4 cell.

**Table 1 membranes-12-00408-t001:** The elemental composition of the snail shells and HAn of EDX spectra.

Element (%)	Snail Shell *Atactodea glabrata*	HAn *Atactodea glabrata*
Ca	35.54	24.53
P	0.01	20.67
O	61.24	48.48

**Table 2 membranes-12-00408-t002:** Antimicrobial activities of snail shell powder *Atactodea glabrata* against different pathogenic microorganisms.

Tested Organisms	Inhibition Zone Diameter (mm)	
	Snail Shell (*Atactodea glabrata*)	Standard
**Gram-positive bacteria**		Gentamicin
*Staphylococcus aureus (ATCC 538)*	20 ± 0.06	22 ± 0.03
*Bacillus subtilis (ATCC 6633)*	32 ± 0.12	26 ± 0.09
**Gram-negative bacteria**		Gentamicin
*Klebsiella pneumoniae (ATCC 13883)*	No activity	25 ± 0.15
*Escherichia coli (ATCC 8739)*	26 ± 0.07	30 ± 0.16
**Fungi**		Amphotericin B
*Candida albicans (ATCC 10221)*	17 ± 0.04	21 ± 0.06
*Aspergillus niger*	No activity	15 ± 0.08

**Table 3 membranes-12-00408-t003:** Antimicrobial activities of HAn prepared from snail shell powder of *Atactodea glabrata* against different pathogenic microorganisms.

Tested Organisms	Inhibition Zone Diameter (mm)	
	HAn	Standard
**Gram-positive bacteria**		Gentamicin
*Staphylococcus aureus (ATCC 6538)*	30 ± 0.03	22 ± 0.03
*Bacillus subtilis (ATCC 6633)*	43 ±0.05	26 ± 0.09
**Gram-negative bacteria**		Gentamicin
*Klebsiella pneumoniae (ATCC 13883)*	42 ± 0.14	25 ± 0.15
*Escherichia coli (ATCC 8739)*	19 ± 0.18	30 ± 0.16
**Fungi**		Amphotericin B
*Candida albicans (ATCC 10221)*	44 ± 0.07	21 ± 0.06
*Aspergillus niger*	22 ± 0.19	15 ± 0.08

**Table 4 membranes-12-00408-t004:** Minimum Inhibitory Concentration (MIC) of HAn by (µg/mL).

Pathogenic Microorganism	HAn
*Staphylococcus aureus (ATCC 6538)*	7.8 ± 1.24
*Escherichia coli (ATCC 8739)*	25 ± 0.85
*Bacillus sutilis (ATCC 6633)*	0.97 ± 1.35
*Klebsiella pneumoniae (ATCC 13883)*	3.9 ± 1.46
*Candida albicans (ATCC 10221)*	0.97 ± 0.76

## Data Availability

Data are available on reasonable request from the authors.

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
