# Peer review of "Synthesis of Natural Nano-Hydroxyapatite from Snail Shells and Its Biological Activity: Antimicrobial, Antibiofilm, and Biocompatibility"

_membranes, 2022, doi:10.3390/membranes12040408_

Round 1

Reviewer 1 Report

In this manuscript the authors present the obtaining of hydroxyapatite nanoparticles from snail shells, waste material. In principle, the nail shells can be used as precursors in the production of calcium phosphate. It is interesting to be able to recycle this waste product through a biomedical application that overcomes resistance to antibiotics and biocompatibility with normal tissues.

This is a very interesting manuscript, but some points should be addressed before the manuscript be considered for publication. These are the following:

1). Page 7, first paragraph. It is indicating the figure 4, but according to what is indicated in the context, this figure was not included.

2). Table 1 and text. Are the EDS percentages values in at% or wt%? Indicate and comment.

3). Page 8, TEM section (and conclusions). The particles are said to be spherical with diameters of 13 and 15 nm. From the images presented in figure 4, it is difficult to conclude these statements. Please improve Figure 4 to present better TEM images to support the comments. It is also recommended in figure 4 to include the corresponding dark field images and the diffraction pattern that allows establishing that these particles correspond to hydroxyapatite nanoparticles.

Author Response

1

In this manuscript the authors present the obtaining of hydroxyapatite nanoparticles from snail shells, waste material. In principle, the nail shells can be used as precursors in the production of calcium phosphate. It is interesting to be able to recycle this waste product through a biomedical application that overcomes resistance to antibiotics and biocompatibility with normal tissues.

This is a very interesting manuscript, but some points should be addressed before the manuscript be considered for publication. These are the following:

1). Page 7, first paragraph. It is indicating the figure 4, but according to what is indicated in the context, this figure was not included.

Response: 

Thank you so much for your observations. There is an error in the first paragraph indicating (Table.1 )but not (figure. 4).

2). Table 1 and text. Are the EDS percentages values in at% or wt%? Indicate and comment.

Response: 

The percentages of Ca, O, and P in at %. To determine the ratio of each element to the other

3). Page 8, TEM section (and conclusions). The particles are said to be spherical with diameters of 13 and 15 nm. From the images presented in figure 4, it is difficult to conclude these statements. Please improve Figure 4 to present better TEM images to support the comments. It is also recommended in figure 4 to include the corresponding dark field images and the diffraction pattern that allows establishing that these particles correspond to hydroxyapatite nanoparticles.

Response: 

Revised as requested. We have written the discussion part and added more scientific discussion. The better TEM images have been added.

Reviewer 2 Report

In my humble opinion, this manuscript is of a moderate scientific quality and might be published as is if a room is available. There are many papers on hydroxyapatite production from various types of shells and I see nothing new, nor original in this study, if compared to the previously published papers on the same topic.

Author Response

2

In my humble opinion, this manuscript is of a moderate scientific quality and might be published as is if a room is available. There are many papers on hydroxyapatite production from various types of shells and I see nothing new, nor original in this study, if compared to the previously published papers on the same topic.

Response:

Thank you for your comments which will help us improve the quality of our manuscript. According to previous studies, biofilm is an essential driver of antibiotic-resistant bacteria. The high resistance of biofilms to current antimicrobial treatments is by all accounts challenge. So, developing new antimicrobial therapies that can overcome this resistance is urgent. The results of our study showed that the prepared HAn from the snail shell Atactodea glabrata presented a higher inhibitory effect than the standard compounds against all tested organisms. Also, the HAn displayed potent antibiofilm against tested organisms. As well as the biocompatibility against normal cells. So,  The prepared HAn is promising for producing antimicrobial and anti-biofilm agents to overcome multidrug-resistance bacteriain numerous biomedical applications, including dentistry.

Reviewer 3 Report

The article entitled: Synthesis of Natural Nano-hydroxyapatite from Snail Shells and its Biological Activity: Antimicrobial, Antibiofilm, and Biocompatibility describes production of calcium phosphate-based material from snail shells. The great part of the manuscript deals with biological studies including i.e. antibiofilm activity and MTT tests of materials. However, in my opinion, the material part with respect to nHA characteristics has not been well considered and discussed. I have a few comments:

1.In the section 2.3. the preparation of snail powder- the Authors clean and dried the snails(?) in the oven at 100°C, or rather dried the snail shells?

2.The number of JCPDS card should be given in the section 2.5.2.

3.The authors should explain why they used the HSF-4 cell line.

4.The positions of the FT-IR bands in the text of the manuscript and in Figure 1 are not the same. For example, the Authors are describing bands at 1471.81 cm-1, 857.87 cm-1 and 712.5cm-1 and in the Figure 1a we can see these bands at 1448.15 cm-1, 880.60 cm-1 and 712.66cm-1.

5.The intensive bands at 1100, 1167, 1137 and 1155 cm-1  were not described (I believe these bands are not typical for hydroxyapatite).

6.XRD studies – usually on hydroxyapatite diffractograms we observe the most intense XRD reflexes at 2theta between 30-35°C . In this study another intense reflexes, not typical for hydroxyapatite, were also observed. How do the Author explain the existence of those reflexes? How can the Authors assign them?

7.EDX study. Stoichiometric hydroxyapatite has the Ca/P~1.67. The results of EDX analysis show that for the material produced from snail shells Ca/P is equal 0.5. The hydroxyapatite can be non-stoichiometric but this Ca/P ratio is too low to indicate the presence of hydroxyapatite.

8.Generally, I am not convinced that the material composed solely of nHA. In my opinion the results of FT-, XRD and EDX studies do not confirm the presence of only one phase. Rather the mixture of few phases, that contains also hydroxyapatite.

9.In Figure 1 the axis subscriptions are too small.

10.Figure 3 is doubled.

11.The axis should be described as 2 theta (°).

12.The introduction section should be improved. At least short introduction to nano-hydroxyapatite based materials and their possible applications and/or antibacterial action should be given by the Authors.

In my opinion without thorough reflection and analysis of the phase composition, the article is not suitable for publication. We cannot discuss the influence of material on biological systems without the knowledge about its composition.

Author Response

3

The article entitled: Synthesis of Natural Nano-hydroxyapatite from Snail Shells and its Biological Activity: Antimicrobial, Antibiofilm, and Biocompatibility describes production of calcium phosphate-based material from snail shells. The great part of the manuscript deals with biological studies including i.e. antibiofilm activity and MTT tests of materials. However, in my opinion, the material part with respect to nHA characteristics has not been well considered and discussed. I have a few comments:

1.In the section 2.3. the preparation of snail powder- the Authors clean and dried the snails(?) in the oven at 100°C, or rather dried the snail shells?

Response: 

To clean and dry the snails.

2.The number of JCPDS card should be given in the section 2.5.2.

Response: 

The number of JCPDS card have been added.

3.The authors should explain why they used the HSF-4 cell line.

Response: 

According to ISO 10993, Part 5: Tests for in vitro cytotoxicity, describes test methods to assess the in vitro cytotoxicity of medical materials. These methods are designed to determine the biological response of any mammalian cells in vitro using appropriate biological parameters. So, we used the normal human fibroblast cell line HSF-4.

4.The positions of the FT-IR bands in the text of the manuscript and in Figure 1 are not the same. For example, the Authors are describing bands at 1471.81 cm-1, 857.87 cm-1 and 712.5cm-1 and in the Figure 1a we can see these bands at 1448.15 cm-1, 880.60 cm-1 and 712.66cm-1.

Response: 

Thank you for your careful review and comments, which will help us improve the quality of our manuscript. Revised as requested, this part was rewritten, and we added more scientific discussion.

5.The intensive bands at 1100, 1167, 1137 and 1155 cm-1  were not described (I believe these bands are not typical for hydroxyapatite).

Response: 

Revised as requested, this part was rewritten, and we added more scientific discussion.

6.XRD studies – usually on hydroxyapatite diffractograms we observe the most intense XRD reflexes at 2theta between 30-35°C. In this study another intense reflexes, not typical for hydroxyapatite, were also observed. How do the Author explain the existence of those reflexes? How can the Authors assign them?

Response: 

Revised as requested, the XRD pattern showed that the sample was mainly constituted of Hydroxyapatite mixed with a phase of tricalcium phosphate.

7.EDX study. Stoichiometric hydroxyapatite has the Ca/P~1.67. The results of EDX analysis show that for the material produced from snail shells Ca/P is equal 0.5. The hydroxyapatite can be non-stoichiometric but this Ca/P ratio is too low to indicate the presence of hydroxyapatite.

Response: 

There is an error. The elements were replaced by each other.a Ca/P ratio close to 1.19. The error has been corrected.

8.Generally, I am not convinced that the material composed solely of nHA. In my opinion the results of FT-, XRD and EDX studies do not confirm the presence of only one phase. Rather the mixture of few phases, that contains also hydroxyapatite.

Response: 

Thank you for your comments which will help us improve the quality of our manuscript. There is a mistake in the EDX result a Ca/P ratio close to 1.19. Also, the XRD pattern showed that the sample was mainly constituted of Hydroxyapatite mixed with a phase of tricalcium phosphate. The result of FTIR was rewritten, and we added more scientific discussion

9.In Figure 1 the axis subscriptions are too small.

Response: 

Revised as requested, the axis subscriptions wasrewritten.

10.Figure 3 is doubled.

Response: 

Thank you for your observation, and we deleted the doubled figure

11.The axis should be described as 2 theta (°). Θ o°

Response: 

Revised as requested.

12.The introduction section should be improved. At least short introduction to nano-hydroxyapatite based materials and their possible applications and/or antibacterial action should be given by the Authors.

Response: 

Thank you for your comment.The introduction section was revised as requested;the introduction section was rewritten, Line 77 - 111.

In my opinion without thorough reflection and analysis of the phase composition, the article is not suitable for publication. We cannot discuss the influence of material on biological systems without the knowledge about its composition.

Reviewer 4 Report

The manuscript deals with the preparation of hydroxyapatite from snail shells using a route based on calcination followed by chemical treatment and sintering. The topic is relevant and interesting, considering the large number of snail shells that are currently discharged as waste materials. The manuscript advocates for the valorization of Atactodea glabrata shells and also discusses published outcomes related to hydroxyapatite prepared from other natural sources

In my opinion, this manuscript can be improved by addressing the following observations and suggestions:

  • Line 66 – ref to. "When crustaceans are eaten much calcium- and HA-rich waste is produced”. The affirmation is correct but please provide an introductory description of hydroxyapatite somewhere before this phrase;
  • Line 73 - please provide references for the production of hydroxyapatite from plants.
  • Lines 195-198 - please check and edit the two phrases for more clarity (i.e. check the grammar and describe the relationship between FTIR results, aragonite, and sulfate);
  • Lines 234-235 - The “prior finding” mentioned here is not referenced. Also, the vast majority of dedicated literature agrees that hydroxyapatite has a Ca/P ratio close to 1.67. Please discuss the low Ca/P ratio in your samples;
  • Lines 257-259 - Please discuss the uncertainty related to the inhibitory effect of snail shell powder against B. subtilis, given the standard deviations of both sample and control (if one considers the standard deviation, it is difficult to affirm that snail shell powder has a higher inhibitory effect);
  • Line 294 - 295 – Please enhance the arguments for the following affirmation “The highest activity of the extracted HAn can be explained by selecting the best calcination temperature” given that all samples were calcined at the same temperature. Please discuss the rationale behind considering 900°C as the optimal calcination temperature (ideally accompanied by thermal analysis results).
  • Figure 3 - There are two XRD results presented, both named hydroxyapatite. Please write which is the one for HAn from snail shells and what sample was analyzed to obtain the other result. Also, it would be very useful to mark the peaks corresponding to hydroxyapatite on your XRD results, to rapidly identify if there are other trace phases in the samples.
  • Figure 4 - The magnification and other information written under the TEM images are not readable. Please edit the image for increasing the font of the text.

Author Response

4

The manuscript deals with the preparation of hydroxyapatite from snail shells using a route based on calcination followed by chemical treatment and sintering. The topic is relevant and interesting, considering the large number of snail shells that are currently discharged as waste materials. The manuscript advocates for the valorization of Atactodea glabrata shells and also discusses published outcomes related to hydroxyapatite prepared from other natural sources

In my opinion, this manuscript can be improved by addressing the following observations and suggestions:

Line 66 – ref to. "When crustaceans are eaten much calcium- and HA-rich waste is produced”. The affirmation is correct but please provide an introductory description of hydroxyapatite somewhere before this phrase;

Response: 

Thank you for your observation, and we added an introductory description of hydroxyapatite in Line 66-70 and lines 77 to111.

Line 73 - please provide references for the production of hydroxyapatite from plants.

Response: 

Revised as requested, and we added references for the production of hydroxyapatite from plants.

Lines 195-198 - please check and edit the two phrases for more clarity (i.e. check the grammar and describe the relationship between FTIR results, aragonite, and sulfate);

Response: 

Revised as requested

  • Lines 234-235 - The “prior finding” mentioned here is not referenced. Also, the vast majority of dedicated literature agrees that hydroxyapatite has a Ca/P ratio close to 1.67. Please discuss the low Ca/P ratio in your samples;

Response: 

There is an error. The elements were replaced by each other.a Ca/P ratio close to 1.19. The error has been corrected.

Lines 257-259 - Please discuss the uncertainty related to the inhibitory effect of snail shell powder against B. subtilis, given the standard deviations of both sample and control (if one considers the standard deviation, it is difficult to affirm that snail shell powder has a higher inhibitory effect);

Response:

The standard deviations of snail shells are about ±0.12 and ±0.09 for gentamicin. Also, to affirm that snail shell powder has a higher inhibitory effect, we added a photo for the inhibition zone of both sample and control against B. subtilis (Figure. 5).

  • Line 294 - 295 – Please enhance the arguments for the following affirmation “The highest activity of the extracted HAn can be explained by selecting the best calcination temperature” given that all samples were calcined at the same temperature. Please discuss the rationale behind considering 900°C as the optimal calcination temperature (ideally accompanied by thermal analysis results).

Response: 

The statement has been modified line 365 to 369.

The temperature at 900 °C is not optimal for all samples, but according to the previous studies, commercially available HAs possessed the heat treatment estimated to be between 900°C and 1300°C and this agreement with our results in which the sample was prepared at 900°C. Thus, the presence of slightly basic compounds (HA, TCP) neutralizes the acid molecules, provides a weak pH-buffering effect at the polymer surface and, therefore, reduces the bacterial growth in which bacteria create acid in biofilms on dental surfaces

  • Figure 3 - There are two XRD results presented, both named hydroxyapatites. Please write which is the one for HAn from snail shells and what sample was analyzed to obtain the other result. Also, it would be very useful to mark the peaks corresponding to hydroxyapatite on your XRD results to rapidly identify if there are other trace phases in the samples.

Response: 

Thank you for your observation, and we deleted the doubled figure. There is only one figure for XRD. We made a mark on the peak corresponding to hydroxyapatite.

  • Figure 4 - The magnification and other information written under the TEM images are not readable. Please edit the image for increasing the font of the text.

Response: 

 Revised as requested. We have written the discussion part and added more scientific discussion. The better TEM images have been added.

Round 2

Reviewer 1 Report

Not further comments.

Reviewer 2 Report

A revised version of the manuscript looks a bit better and, unless other reviewers find additional imperfections, in my humble opinion, this version might be published as is.

Reviewer 3 Report

Introduction

 1. “Emerging bioceramics, which are widely employed in numerous biomedical applications,
including dentistry, are biologically relevant forms of CaPO4.”- The chemical formula CaPO4 is incorrect. I suggest to use abbreviation CaPs for calcium phosphates.

2. “Because their composition and structure are similar to those of humans, they behave similarly to human teeth and bones; as a result, CaPO4 has extraordinary properties.” Please rephrase this sentence( a lot of repetition in one sentence) . Please decide and use one - singular or plural form (they.., CaPO4 has…).

3. “Following that, the infiltrated collagen matrix of dentin could provide a suitable scaffold for dentin remineralization, with the infiltrated HA particles acting as seeds within the collage matrix and given the appropriate remineralizing agents.” Please rephrase this sentence (a lot of repetition in one sentence).

4. “Coatings of CaPO4 (both HA and -TCP have been effectively applied to titanium implants, and the coated implants have performed well were discovered to be suitable for use as anchoring in short-term orthodontics.” The bracket is missing.

5.  “New bone growth stimulators [25].” I have the impression that the part of this sentence is missing(?).

6. XRD analysis” hydroxyapatite was done by powder X-ray diffraction.” The part of the sentence is missing?

Results

7. The general comment. FT-IR study - the Authors describe absorption bands from beta TCP. Furthermore, the Authors write that XRD studies revealed presence of tricalcium phosphate (beta form?). However, in Fig.3 only reflexes from hydroxyapatite are marked. Also only one JCPDS card is described. Thus, the reader may be confused about the exact phase composition of the final material. I possible, the quantitative XRD study would be helpful (Rietveled refinement).

Author Response

comments and Suggestions for Authors

Introduction

  1. “Emerging bioceramics, which are widely employed in numerous biomedical applications,
    including dentistry, are biologically relevant forms of CaPO4.”- The chemical formula CaPO4 is incorrect. I suggest to use abbreviation CaPs for calcium phosphates.

Response: 

Thank you for your comments which will help us improve the quality of our manuscript. Revised as requested.

  1. “Because their composition and structure are similar to those of humans, they behave similarly to human teeth and bones; as a result, CaPO4 has extraordinary properties.” Please rephrase this sentence( a lot of repetition in one sentence) . Please decide and use one - singular or plural form (they.., CaPO4 has…).

Response:

Thank you for your comments. The sentence has been rephrased.

  1. “Following that, the infiltrated collagen matrix of dentin could provide a suitable scaffold for dentin remineralization, with the infiltrated HA particles acting as seeds within the collage matrix and given the appropriate remineralizing agents.” Please rephrase this sentence (a lot of repetition in one sentence).

Response:

Thank you for your comments. The sentence has been rephrased.

  1. “Coatings of CaPO4 (both HA and -TCP have been effectively applied to titanium implants, and the coated implants have performed well were discovered to be suitable for use as anchoring in short-term orthodontics.” The bracket is missing.

Response:

Revised as requested.

  1. “New bone growth stimulators [25].” I have the impression that the part of this sentence is missing(?).

Response:

Revised as requested.

  1. XRD analysis” hydroxyapatite was done by powder X-ray diffraction.” The part of the sentence is missing?

Response:

Thank you for your observation. Revised as requested.

Results

  1. The general comment. FT-IR study - the Authors describe absorption bands from beta TCP. Furthermore, the Authors write that XRD studies revealed presence of tricalcium phosphate (beta form?). However, in Fig.3 only reflexes from hydroxyapatite are marked. Also only one JCPDS card is described. Thus, the reader may be confused about the exact phase composition of the final material. I possible, the quantitative XRD study would be helpful (Rietveled refinement).

Response: 

Thank you for your observation. We made a mark on the peak corresponding to tricalcium phosphate.

Reviewer 4 Report

The authors addressed all my previous queries. Please edit the current form of the manuscript because the figures are misplaced and there are some formatting issues.